Effects of ocean acidification on the dissolution rates of reef-coral skeletons

van Woesik Robert 1 rvw@fit.edu
van Woesik Kelly 2
van Woesik Liana 2
van Woesik Sandra 2
1 Florida Institute of Technology , Melbourne, FL , USA
2 Melbourne, FL , USA
Medina Mónica
Electronic publication date: 2013 Nov 21
Publication date: 2013
Volume: 1
Electronic Location ID: e208
Received 2013 Aug 26; Accepted 2013 Oct 22
Copyright: © 2013 van Woesik et al.
Copyright year: 2013
Copyright holder: van Woesik et al.
License: This is an open access article distributed under the terms of the Creative Commons Attribution License, which permits unrestricted use, distribution, and reproduction in any medium, provided the original author and source are credited.
License URL: https://creativecommons.org/licenses/by/3.0/

Keywords: Corals, Ocean acidication, Coral reefs, Reef growth, Sea-level rise, Climate

Funding: There was no funding for this work.

==============================
Ocean acidification threatens the foundation of tropical coral reefs. This study investigated three aspects of ocean acidification: (i) the rates at which perforate and imperforate coral-colony skeletons passively dissolve when pH is 7.8, which is predicted to occur globally by 2100, (ii) the rates of passive dissolution of corals with respect to coral-colony surface areas, and (iii) the comparative rates of a vertical reef-growth model, incorporating passive dissolution rates, and predicted sea-level rise. By 2100, when the ocean pH is expected to be 7.8, perforate Montipora coral skeletons will lose on average 15 kg CaCO3 m−2 y−1, which is approximately −10.5 mm of vertical reduction of reef framework per year. This rate of passive dissolution is higher than the average rate of reef growth over the last several millennia and suggests that reefs composed of perforate Montipora coral skeletons will have trouble keeping up with sea-level rise under ocean acidification. Reefs composed of primarily imperforate coral skeletons will not likely dissolve as rapidly, but our model shows they will also have trouble keeping up with sea-level rise by 2050.

Introduction

Ocean acidification

As humans continue to burn fossil fuels at an unprecedented rate, the concentration of carbon dioxide (CO2) in the atmosphere is presently higher than it has been for the last 420,000 years (Petit et al., 1999; Hansen et al., 2006; Hoegh-Guldberg et al., 2007). The oceans uptake a large proportion of the atmospheric CO2, increasing the concentrations of both carbonic acid and bicarbonate ions, and reducing the concentration of carbonate ions, shifting the ocean’s acid–base balance toward a lower pH (Broecker, 1983; Caldeira & Wickett, 2003; Silverman et al., 2009). The increase in ocean acidification directly threatens calcifying marine organisms, such as reef-building corals and the myriad of species that rely on corals for protection and sustenance (Hoegh-Guldberg et al., 2007; Rodolfo-Metalpa et al., 2011).

Ocean pH has already decreased by 0.1 pH units since the 18th century, and is expected to drop by another 0.2–0.4 pH units by 2100. Yet the oceans are not homogenous in regard to rates of reductions in carbonate ions. Although warm waters increase reaction rates, thermodynamic principles and Henry’s Law tells us that cool temperate and polar waters absorb asymmetrically more CO2 than tropical waters, and are therefore closer to unity than the more super-saturated tropical waters (Broecker, 1983). Yet the tropical oceans are changing at a more rapid rate and are acidifying more quickly than the cooler waters most likely because of the relationship of rapidly increasing ocean temperature and reaction rates (Zeebe et al., 2008). Moreover, the Pacific Ocean is more acidic than the Atlantic Ocean, and shoaling saturation depth is around 500 m in the Pacific and 4500 m in the Atlantic (Feely et al., 2004; Millero, 2007).

There is increasing evidence that ocean acidification, through the increase in the partial pressure of carbon dioxide (pCO2) and the subsequent changes in the concentration of carbonate and bicarbonate ions, reduces rates of coral calcification, which are directly proportional to the saturation state of aragonite in the shallow oceans (Langdon & Atkinson, 2005). Other studies have shown that calcification rates are proportional to the concentration of carbonate ions in the water column (Anthony et al., 2008; Marubini et al., 2008). These studies are essentially synonymous, however, because the aragonite and calcite saturation state (Ω) is the product of the concentrations of calcium and carbonate ions divided by an equilibrium constant. Since the salt concentration, including calcium ions, stemming from terrestrial weathering hasn’t changed in the oceans for over 1.5 billion years, the aragonite saturation state is essentially a measure of carbonate ions in the oceans.

Perhaps more importantly is the strong interaction effects between temperature and ocean acidification on coral calcification rates (Reynaud et al., 2003; Erez et al., 2010). Indeed, the optimal window of physiological performance of a given marine species at a given temperature will be narrowed under acidification (Portner, 2010). Calcification of corals under ambient temperature do not necessarily change with increased pCO2, but calcification decreases when both temperature and pCO2 are elevated (Reynaud et al., 2003). Yet several studies have shown that many corals are unaffected by external carbonate ion concentrations because they have the capacity to up-regulate internal pH by actively exchanging internal hydrogen ions for calcium ions through Ca-ATPase transportation (Al-Horani, Al-Moghrabi & de Beer, 2003; Allemand et al., 2004; McCulloch et al., 2012). By modifying their internal chemistry, live corals may buffer themselves from ocean acidification. Coral skeletons, however, have no internal-buffering capacity because they are not protected by coral membranes (Rodolfo-Metalpa et al., 2011; Ries, 2011). Coral skeletons are instead subjected to the raw and immediate threats of ocean acidification and will be subjected to dissolution when the ocean’s pH declines.

Accretion of coral reefs

The accretion of coral reefs occurs over geological time periods when rates of calcium carbonate (CaCO3) production exceed rates of destruction and dissolution (Neumann & MacIntyre, 1985; Buddemeier & Hopley, 1988; Glynn, 1997; Perry et al., 2013). The interaction between production and destruction depends on the consistency of coral cover through time. For example, where coral cover is consistently low, reef accretion is minimal (Neumann & MacIntyre, 1985). Most modern reefs, however, support little more than 28% live coral cover (Bruno & Selig, 2007), and are essentially veneers over pre-existing, antecedent foundations of CaCO3 (Adey, 1978; Hopley, 1982). For example, the Florida Keys only supported, on average, 2–3% of live coral cover in 2011 (Office of National Marine Sanctuaries, 2011). Therefore, reefs with high carbonate cover and few live corals are particularly vulnerable to ocean acidification.

The average modern, shallow seaward coral reef in the Indo-Pacific, with high coral cover, has been estimated to produce about 4 kilograms of calcium carbonate per square meter of reef per year, which equates with an upward reef-growth rate of approximately 3 mm y−1 (Smith & Kinsey, 1976). These estimates were based on alkalinity reduction techniques subjected to a pH of 8.2, equivalent to the pH of today’s oceans. By 2100 the ocean’s pH is expected to be 7.8, and we hypothesize that the destructive processes associated with ocean acidification might outweigh the constructive processes. The rates of dissolution of reef framework may, however, also depend on flow rates, the extent of cementation of reef framework, and on the porosity of corals and their surface area.

Reef cementation and coral porosity

Reefs vary in porosity depending on both: (i) the local rates of sedimentation and the extent to which those sediments become consolidated, or lithified, within the reef framework, and (ii) the extent of cementation of the reef framework. Both processes depend in part on exposure to water-flow rates (MacIntyre & Marshall, 1988). High-energy, windward reefs consistently exposed to large waves are generally more highly cemented than low-wave energy, leeward reefs because mass-transfer rates influence rates of cementation. Cementation involves the infilling of intra-skeletal pores with either Mg calcite or aragonite (MacIntyre & Marshall, 1988). While the extent of cementation affects the dislodgment of reef substrate and the tenacity of corals to remain attached to reefs during storms (Madin, Hughes & Connolly, 2012), the extent of reef cementation may also affect dissolution rates during ocean acidification because the infilling of pores by cements decreases the surface area of exposure (Cubillas et al., 2005).

Reef corals also vary in porosity (Gladfelter, 1982; Hughes, 1987). Although all modern corals secrete orthorhombic aragonite fusiform crystals, as small as 1–3 µm (Gladfelter, 1982), corals vary considerably in the arrangement of the crystals, which influences the internal surface area that is exposed (Fig. 1). Fast-growing corals, such as Montipora and Acropora, are mostly perforate corals (Gladfelter, 1982), whereas slow-growing corals, such as Pectinia and Symphyllia, are imperforate (Table 1). An extreme example of imperforate skeletons is evident by the observation of occasional floating, massive Symphyllia colonies (DeVantier, 1992). Because of the fused nature of the dissepiments and their imperforate skeletons, gases are trapped in the septal chambers and upon dislodgement from reefs, for example during a storm, the colonies will float. Perforate corals, however, do not have the capacity to isolate septal chambers.

Table 1 Porosity of scleractinian corals.

Scleractinian coral families, the number of species in each family, and the general porosity of the coral skeletons. Classifications were based on the porosity of the colony walls, the coenosteum, and the collumellae at the scale of 1 mm2. There are approximately 404 perforate species and 432 imperforate coral species, globally.

Family	Number of species	Porosity	
Acroporidae	271	Perforate	
Agariciidae	45	Imperforate	
Astrocoeniidae	15	Imperforate	
Caryophylliidae	7	Imperforate	
Dendrophylliidae	19	Imperforate	
Euphyllidae	17	Imperforate	
Faviidae	130	Imperforate	
Fungiidae	46	Imperforate	
Meandrinidae	12	Imperforate	
Merulinidae	12	Imperforate	
Mussidae	52	Imperforate	
Oculinidae	16	Imperforate	
Pectiniidae	29	Imperforate	
Pocilloporidae	31	Imperforate	
Poritidae	101	Perforate	
Siderastreidae	32	Perforate	
Trachphylliidae	1	Imperforate	

Figure 1 Montipora.

Scanning electron microscope image of Montipora skeleton; scale bar is 500 µm.

The internal porosity of coral skeletons, at the scale of 0.5–1 mm (Fig. 1), increases the available surface area of chemical exchange and therefore increases the potential rates of dissolution. Walter & Morse (1984) showed that rates of dissolution of skeletal carbonates were inversely related to grain-size diameter and surface roughness, with fine grained carbonates dissolving faster than large, rough surfaces. However, we should not discount the possibility that perforate and imperforate corals also differ in other aspects, beyond the obvious differences in porosity, and therefore we question whether surface area is a useful predictor of rates of passive dissolution of both perforate and imperforate corals.

This study will examine whether the porosity and the surface area of coral skeletons will influence their rate of dissolution when the ocean pH is 7.8, which is predicted to occur by 2100. More specifically, we tested three hypotheses: (1) that perforate Montipora coral skeletons are more likely to passively dissolve than imperforate Pectinia coral skeletons at a pH of 7.8, (2) that the rates of passive dissolution of coral-colony skeletons are proportional to their surface areas, and (3) future reef accretion rates under ocean acidification will differ depending on the nature of the coral assemblages, with perforate coral assemblages unable to keep up with predicted sea-level rise and imperforate coral assemblages faring a better chance at keeping up with sea-level rise and ocean acidification.

Materials and Methods

Acidification experiments

In order to test the first hypothesis, perforate Montipora colonies (Fig. 1) and imperforate Pectinia colonies without tissue (Fig. 2) were used to make comparisons of weight loss when immersed in seawater and held in zero-flow conditions (i.e., to test passive dissolution) at a pH of 8.2, equivalent to the pH of today’s oceans, and compared with colonies held at a pH of 7.8, which is predicted to occur globally by 2100. Fifteen skeletal samples (≤5 cm) of Montipora spp. colonies and fifteen skeletal samples of Pectinia spp. were collected from the fringing reefs of Okinawa, Japan in 2001. In order to test the second hypothesis, we used a variety of growth forms of Montipora, including submassive, branching, encrusting, and foliose. Colonies of Pectinia with different surface areas were used for experimental treatments, but all samples were foliose because Pectinia is only found as foliose colonies on modern coral reefs.

Figure 2 Pectinia.

Scanning electron microscope image of Pectinia skeleton; scale bar is 500 µm.

Before pH treatments, the samples were placed in a drying oven at 40°C for 48 h and weighed (g) using a Sartorius Research Balance. Each treatment sample was then placed in a separate container of seawater that was maintained at a pH of 7.8 by adding diluted acetic acid to match the predicted pH of the seawater in the year 2100 (Intergovernmental Panel on Climate Change (IPCC, 2007)). The control samples were placed in seawater that was maintained at a pH of 8.2, to match modern ocean conditions, and maintained at 24°C and a salinity of 35. Total alkalinity was not measured in this study. Seawater was changed every 2 days. After 7 days the samples were rinsed and dried in a drying oven at 40°C for 48 h, and re-weighed. The volume of each coral sample (mL) was calculated using a displacement method and the surface area of each coral sample (cm2) was calculated using a single wax-dipping method (Veal, Carmi & Fine, 2010).

Data analyses

The difference in dry weight (g) before and after the acid treatment was calculated for each coral sample. To correct for differences in initial weight, the loss of calcium carbonate was divided by each coral’s initial weight. To compare differences in dissolution rates that may have varied in accordance with growth form, we undertook an analysis of variance (ANOVA) and a Tukey’s post-hoc test using R (R Development Core Team, 2012). The relationship between the surface area, volume, and the loss of calcium carbonate was examined using curve fitting with Matlab®.

Accretion-dissolution model

The loss of calcium carbonate was extrapolated from the change in calcium carbonate per gram cm−2 d−1, to the equivalent loss of calcium carbonate per kg m−2 y−1. This loss was compared with the geological literature and converted to the approximate equivalent of vertical reduction of reef framework in mm per year (Smith & Kinsey, 1976). The loss was compared with predicted sea-level rise (Vermeer & Rahmstorf, 2009). In order to achieve this goal, the reef accretion rates were modeled as an ordinary differential equation: (1) dA/dt=(a.A)/A+b.S−(c.D)/A,

where A is the accretion of a reef relative to time (t); a is the accretion coefficient determined by coral and coralline algal growth minus the bioerosion rates (input as 7 mm y−1 for reefs that accrete the maximum of 10 kg CaCO3 m−2 y−1; 3 mm y−1 for reefs that accrete 4 kg CaCO3 m−2 y−1; and 0.75 mm y−1 for reefs that accrete 1 kg CaCO3 m−2 y−1, with a 50% average reef porosity, after Kinsey, 1979; Smith, 1983); b is a coefficient for sedimentation (S), input as 1 mm per year for consistency; and c is a coefficient for the dissolution (D) rates. The equations were solved using Runge–Kutta methods using the ode45 solver in Matlab® (code is available in the Appendix S1).

The results of passive dissolution were input into our reef-growth model and compared with projections of global sea-level rise, from 1990 to 2100 following Vermeer & Rahmstorf (2009), which did not consider regional isostatic rebound effects, regional tectonics, and local land-use effects. The sea-level rise projections used different IPCC (2007) emission scenarios, including the B1 scenario representing a +1.8°C global increase in temperature, the A2 scenario representing a +3.4°C global increase in temperature, and the A1F1 scenario representing a 4°C global increase in temperature.

Results

There was a significant difference (p < 0.0258) in coral skeleton weight loss that was dependent on coral colony porosity (Fig. 3). The skeletons of foliose, perforate Montipora coral colonies passively dissolved significantly (post-hoc Tukey test, p < 0.011) faster than the skeletons of foliose, imperforate Pectinia coral colonies (Fig. 3). The skeletons of foliose Montipora corals also lost more calcium carbonate than other Montipora growth forms (Fig. 3). Foliose Montipora corals also lost more calcium carbonate than other Montipora growth forms (Fig. 3). There was a strong negative relationship between the surface area of Montipora corals and the loss of calcium carbonate, suggesting that the larger the surface area of Montipora colonies the more rapidly the corals dissolved (Fig. 4). The rate of calcium carbonate loss followed the equation, CaCO3 loss = −0.005 × exp0.017* surface area. The loss of CaCO3 of perforate Montipora was approximately 0.000042 g CaCO3 cm−2 d−1 (−0.42 g CaCO3 m−2 d−1, or −15.3 kg CaCO3 m−2 y−2). This loss in calcium carbonate is approximately equivalent to −10.5 mm of vertical reduction of reef framework per year.

Figure 3 Comparative loss of calcium carbonate.

Loss of calcium carbonate, divided by the initial weight (g), of four different coral growth forms of Montipora coral skeletons, and one growth form of Pectinia coral skeleton, when exposed to pH 7.8 seawater for 7 days. The graph also depicts the controls for Montipora and Pectinia coral skeletons, which were exposed to present-day seawater, at a pH of 8.2, for 7 days. The dashes are the data points, the horizontal lines on each ‘bean’ show the means, and each ‘bean’ shape follows the general distribution of the data relative to density (constructed using the package ‘beanplot’ in R).

In contrast, the skeletons of imperforate Pectinia colonies showed no consistent (passive) dissolution at a pH of 7.8, suggesting that the loss of weight in low pH treatments was no different than the weight loss in controls (Figs. 3 and 5). There was no significant relationship between the surface area of Pectinia coral colonies and their rate of passive dissolution (Fig. 5). There was also no significant relationship between dissolution rates and the volume of either Montipora or Pectinia colonies.

Figure 4 Montipora dissolution.

The relationship between the surface area of the Montipora coral skeletons (cm2) and the loss of calcium carbonate (CaCO3) (g) over 7 days follows the equation loss = −0.005 × exp0.017* surface area. The dots are the data points, the thick, black line represents the equation, and the dotted lines represent the 95% confidence intervals.

Figure 5 Pectinia dissolution.

The relationship between the surface area of the Pectinia coral skeletons (cm2) and the loss of calcium carbonate (CaCO3) (g) over 7 days.

Accretion-dissolution model

The sea-level rise projections from 1990 to 2100 were constructed using different IPCC (2007) emission scenarios, including the B1 scenario, representing a +1.8°C global increase in temperature, the A2 scenario representing a +3.4°C global increase in temperature, and the A1F1 scenario representing a 4°C global increase in temperature (Fig. 6; Vermeer & Rahmstorf, 2009). These sea-level projections were compared with three different reef-building capacities in conjunction with rates of perforate and imperforate coral skeletons (Eq. (1)) under ocean acidification (Figs. 6 and 7). The modeled reef with high dissolution rates, which included perforate skeletons, and consistently high coral cover (10 kg CaCO3 m−2 y−1) is not expected to keep up with sea level rise under ocean acidification (Fig. 6). By contrast, the modeled reef with low dissolution rates, which included imperforate corals, is expected to continue to grow reefs and keep up with sea level rise, but only reefs consistently supporting high coral cover (10 kg CaCO3 m−2 y−1) and only to 2050. Around 2050, the model shows that the rates of sea level rise are expected to increase faster than the rates at which corals can grow reefs (Fig. 7).

Figure 6 Accretion potential of perforate corals and predicted sea-level rise.

The projections of expected rates of coral-reef accretion relative to rates of dissolution of reefs composed of mainly perforate corals, with 3 different densities of corals (low, medium and high modeled as 1, 4, and 10 kg CaCO3 m−2 y−1), along with projections of global sea-level rise (not considering regional isostatic rebound effects, regional tectonics, and local land-use effects) and potential reef-accretion rates from 1990 to 2100 following Vermeer & Rahmstorf (2009) for different IPCC (2007) emission scenarios, where the B1 scenario is green and represents a +1.8°C global increase in temperature; the A2 scenario is blue and represents a +3.4°C global increase in temperature; the A1F1 scenario is red and represents a 4°C global increase in temperature.

Figure 7 Accretion potential of imperforate corals and predicted sea-level rise.

The projections of expected rates of coral-reef accretion relative to rates of dissolution of reefs composed of mainly imperforate corals, with 3 different densities of corals (low, medium and high modeled as 1, 4, and 10 kg CaCO3 m−2 y−1), along with projections of global sea-level rise (as in Fig. 6).

Discussion

This study examined whether the destructive processes involving the dissolution of calcium carbonate might over-ride the accretionary potential of coral reefs when the ocean pH drops to 7.8, which is predicted to occur by 2100. We examined the rates of skeletal dissolution of two Indo-Pacific corals, Montipora and Pectinia, subjected to a pH of 7.8. Rates of passive dissolution were directly proportional to the surface area of corals, but only for the perforate Montipora; dissolution was less predictable for the imperforate Pectinia. The average loss of Montipora CaCO3 per surface area was 15.3 kg m−2 y−1, which was 3 times more than the average growth rates of modern reefs (4 kg CaCO3 m−2 y−1) (Smith, 1983; Smith & Kinsey, 1976; Kinsey, 1979).

We should however, treat the comparative results between skeletal dissolution and reef growth with caution, even though the units match (CaCO3 m−2 y−1). In the comparison above, our data were extrapolated across at least five orders of magnitude spatially, from grams per cm2 to kilograms per m2, and at least six orders of magnitude temporally, from skeletal dissolution over weeks to reef growth over millennia. Yet our results, on the passive dissolution rates of porous Montipora coral skeletons (−0.42 g CaCO3 m−2 d−1) and the recent field results from Cyronak, Santos & Eyre (2013), on the passive dissolution rates of carbonate sediments on Heron Island (Great Barrier Reef, Australia), are the same. Cyronak, Santos & Eyre (2013) also showed that adding flow to experiments more than doubled dissolution rates because of advection processes.

We also note that Smith, Kinsey, and co-workers, originally calculated calcium carbonate production using advection alkalinity reduction techniques that measured change in alkalinity across reef flats over minutes. The maximum rate of modern reef growth has been estimated at 9.6 kg CaCO3 m−2 y−1 on a back-reef of Johnston Atoll (16°N, 169°W), that supported “heavy” coral cover (but the percentage coral cover was not provided in the original publication) (Kinsey, 1979). Other estimates using X-radiographs and extrapolation techniques showed similar results, ranging from 9 kg CaCO3 m−2 y−1 for reefs in the Caribbean with uncharacteristically high coral cover (38%) (Stearn, Scoffin & Martindale, 1977), to less than 1 kg CaCO3 m−2 y−1 for reefs with low coral cover (Dullo, 2005).

Although our results show rapid rates of Montipora dissolution, modern-reef framework is not all Montipora. Rates of carbonate dissolution will also depend on the type of coral assemblages that are present on reefs and their densities. Globally, approximately 404 coral species are perforate, and 432 are imperforate (Table 1), yet most Indo-Pacific reefs are dominated by Acropora, Montipora, Porites, and faviids; and Caribbean reefs are dominated by Porites, Siderastrea, and Orbicella. Therefore, most modern reefs are primarily supporting perforate corals, and these corals have disproportionately contributed to vertical reef accretion through the Holocene (Veron, 1995; Wood, 1999).

Still, changing the pH of seawater is only one of the changes that will occur to reefs subjected to climate change. Sea level will also rise with increasing global temperature (Smith & Buddemeier, 1992; Vermeer & Rahmstorf, 2009). The conservative estimates of sea-level rise from the IPCC (2007), which did not consider ice-sheet dynamics, showed that sea level will increase 20–60 cm by 2100 (approximately 4 mm a year). More recent estimates of sea level rise by Vermeer & Rahmstorf (2009) predict a sea level increase of 75–90 cm by 2100, which is approximately 9 mm a year. Our predictive model, although extrapolating across several spatial and temporal scales, showed that coral reefs composed of perforate skeletons and supporting few live corals, will have trouble keeping up with sea level rise under ocean acidification. These results, although tentative, suggest that more quantitative studies are necessary to determine the potential of reefs to keep up with sea level rise by hierarchically quantifying the production versus dissolution rates of reefs in relation to: (i) coral cover, (ii) coral-community composition, (iii) habitat type, and (iv) regional oceanography.

Supplemental Information

Appendix S1 Appendix

Matlab code for Eq. (1).

Click here for additional data file.

Our special thanks extend to Richard Turner, Biological Sciences Department, College of Science, Florida Institute of Technology, for the use of his laboratory equipment and to Gayle Duncan for preparing the scanning electron microscope images. The corals were collected at (26°21′34.47″ N, 127°44′20.96″ E) with a permit to Robert van Woesik from the Okinawan prefectural government, Japan.

Additional Information and Declarations

Competing Interests

Author Contributions

Field Study Permissions

The authors declare no competing interests.

Robert van Woesik and Kelly van Woesik conceived and designed the experiments, performed the experiments, analyzed the data, contributed reagents/materials/analysis tools, wrote the paper.

Liana van Woesik performed the experiments, contributed reagents/materials/analysis tools, wrote the paper.

Sandra van Woesik conceived and designed the experiments, performed the experiments, contributed reagents/materials/analysis tools, wrote the paper.

The following information was supplied relating to ethical approvals (i.e., approving body and any reference numbers):

The scleractinian corals were collected with a blanket permit from the Okinawan Prefectural Government to Robert van Woesik while he was faculty of the University of the Ryukyus from 1994 to 2001, Okinawa, Japan.

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
