# Peer review of "Effects of ocean acidification on the dissolution rates of reef-coral skeletons"

_PeerJ, doi:10.7717/peerj.208_

## Round 0.1 · original submission · Major Revisions

The three reviewers raise important issues in how the results are presented and discussed. The methodology seems to need better explained as well. Please follow their recommendations that will help you improve your manuscript.

·

Basic reporting

Problems with the figures, see General comments.

Experimental design

Relevance and meaningful problem. Fixable maybe by reformulating hypotheses. See general comments.

Validity of the findings

Conclusion extrapolates. Disconnection problems. Pointless tests of widely accepted phenomenon? Maybe.

Additional comments

Abstract

Line 16: the rates of coral dissolution?

Line 18: rephrase “When the ocean pH is 7.8”

Introduction

Line 26-27: Consider restructuring this sentence.

Line 35: add reference.

Line 37-39: review this sentence. It makes no sense.

Line 45-46 & 59: There seems to be mixed results in the literature about the effects of increased pCO2 on coral calcification rates. If there really is increasing evidence that with increasing pCO2 coral calcification slows down, please add references. I have a problem with the fact that Paragraphs 44-54 and 55-68 seem to contradict one another. I suggest you reformulate or improve the flow between these two.

Line 75: remove or replace “however” with a transition word that fits.

Line 82-83: Are you really hypothesizing that ALL destructive processes will outweigh destructive ones? Please reformulate to match what you are actually testing in your study.

Line 87-96: I must admit I am not an expert on reef porosity, but this paragraph is very confusing. You have to better explain what is lithification and cementation. How are they different? peerJ is not a specialized journal, so I would advise using the same kind of care as if you were submitting to Nature or Science by trying to include as much of your readership as possible by using less jargon and explaining important aspect of your system better and in a simple way.

- What do you refer to when talking about “extent of cementation of the reef framework”?
- Add reference to line 90
- Add reference to line 91. Why is that?
- Line 95: remove repetition.

Line 97: drop “similarly”.

Line 98: move up “fusiform crystals” and add “as small as”. Add references please.

Line 100: “the bulk water”? Are you talking about internal skeletal water? How is that bulky?

Line 103-105: Although this is a real fun fact, it ‘s not necessary here.

Line 107: This is irrelevant to your hypothesis, which comes next. So why is it here?

Line 114-117: I think maybe those questions are not formulated in the best interest of the paper and that is why previous reviewers were ticked off. The problem is that you formulate your hypothesis, about a phenomenon that obeys the laws of physics, which anyone would answer correctly without even conducting any tests. It’s similar to asking: “what is going to dissolve faster between a rock full of holes and a solid rock?”
I suggest you think about why those comparisons are worth pursuing? The expectation that one would have is perforate corals dissolve faster than imperforate ones, right? So why do we need to look at this phenomenon closer?
Could it be that some perforate or imperforate corals don’t use the exact same chemistry to form crystals and therefore tests are necessary to confirm that all perforate corals dissolve at the same rate? I am just brainstorming here, trying to help you getting on the right track. The imperforate corals are porous as well, and must contain some water. That water is similar to ocean water, or is it? Would you expect the trapped water inside imperforate corals to dissolve skeleton from the inside?
The 2nd hypothesis has similar problems to the 1st. The question that everybody will ask you after reading this hypothesis is why would you expect this not to be true? So you really need to make an effort to come up with reasons why this is something worth testing. Do you expect certain corals with higher surface area to dissolve less than corals with lower surface? What brought you to ask those questions?
The 3rd hypothesis is really what this paper should aim to introducing to the community. This is the interesting stuff. We need to get a better understanding of how rates of dissolution differ between different coral substrata/sediment/cementation/lithification/porosity? to be able to accurately predict the effects of ocean acidification on rates of accretion. I suggest you reformulate your intro and your hypotheses to lead to this very relevant and interesting question.

Materials and Methods

Did you remove the tissue before drying samples in the oven? This will affect the rates of dissolution because of differences in skeletal exposure to outside acidified water!

Line 139: please explain your choice of correction. The dissolution will not only be dependent on initial weight/size of the branch, but also porosity, won’t it?

Results

Figure 3: There are 7 coral morphologies and only 4 labels on the x-axis. This axis is not representing a logical value increment; therefore you need to specify the categories. What is the graph showing? I am not familiar with this type of representation. What are those weird shapes and why do they differ?

Figure 4: Why does the y-axis go to -0.04 when the lowest value depicted is only approximately -0.003? How do you explain that the loss of calcium carbonate does not change with increasing surface area?

Line 173-175: I don’t see this at all on Figure 4! I think you swapped Fig 4 and 5.

Figure 6 & 7: You need to get permission to reproduce any results that were previously published by others and cite the permit properly. Legend and axis/label are missing for rate of accretion.

Discussion

First paragraph is out of place. It should be an overview of your main findings. You give information that belongs to the introduction.

Line 198: “Most recent studies” what do you mean?

Line 203-204: this is omitting that calcium carbonate substratum in the ocean is mostly covered by algae or other fouling organisms when live coral is not present.

Line 229-230: 50%? Are you talking about number of species or coral cover? Add reference.

Second paragraph, you state what you have done and give the results again. You don’t discuss them. Put your results in context; explain what they represent and their relevance to natural phenomenon. Help the reader understand the significance of your work.

Third paragraph, this information can help show that your work brings something new to the field and is important. But, the way it is structured, it stands alone without flow or connection to previous or following information. Try to connect your thoughts. Help the reader understand your point by linking your statements.

Forth paragraph, your study is not about coral assemblages. This is out of context, and adds distraction to a discussion already difficult to follow.

Fifth paragraph, line 234-237 needs transition between thoughts or more explanations. Line 244, you mention your results but without making any comparisons to references above. Please discuss your results further in the context of previous knowledge.

Conclusion: First sentence is about results from others! Not a great start for a conclusion. Line 252-254, please rephrase, it’s incomprehensible. Line 255-258, you only studied 1 perforate coral and you are now generalizing for all of them. This is too much of a stretch, please rephrase.

In general, the discussion requires a lot of work. See comments about the introduction and reformulate accordingly.

Reviewer 2 ·

Basic reporting

I have read the manuscript and while I have no doubt that the results presented by the authors support their original and novel hypothesis that perforate corals are more susceptable to ocean acidification then imperforate corals I still have a number of reservations regarding the experimental methodology, which in my opinion should be described in more detail. With respect to the coral skeletons it is not clear if they are devoid entirely of organic matter after the initial and/or final drying treatment during the experiment. It is possible that the weight reduction was caused by metabolism of organic matter with the skeletal remains. Secondly, while the authors state that they added acetic acid (organic acid, which can be metabolised) they do not present any other parameters of the carbonate system. Neither do the authors state how pH measurements were made and in what scale (electrode, litmus paper, activity scale or total hydrogen). Assuming that pH was measured in activity scale then for a pH of 8.2, temperature of 25 deg C, salinity of 35 and total alkalinity of 2250 umol/kg you get a pCO2 of ~370 uatm or 380 ppm at 1 atm dry air and aragonite saturation of 3.4. Assuming that acetic acid was added until pH 7.8 was attained and keeping pCO2 constant at atmospheric equilibrium results in a drastic reduction in total alkalinity, which is very different from seawater. Under these conditions the aragonite saturation will be 0.5, which will be able to cause dissolution of CaCO3. While the pH level projected for 2100 was attained the experimental conditions used do not represent the "natural" conditions that will occur in 2100. Where, assuming that total alkalinity remains at 2250 umol/kg and pH is 7.8 requires an atmospheric PCO2 of ~1000 ppm and results in an aragonite saturation of 1.5, which is above saturation and will not cause dissolution of CaCO3 on its own. Clearly, the experimental conditions do not resemble what may actually occur with increasing atmospheric CO2 and ocean acidification in my opinion. However, it is very likely that aragonite saturation in a reef could be lower than 1 given the additional effects of respiration within carbonates or on the bulk water if the residence time is long enough causing CaCO3 dissolution. I think that the point that the authors have made regarding the susceptibility of perforate as opposed to imperforate corals is valid, but I do not think that they can make the projections that they present in the manuscript because of the problematic experimental system. In conclusion I think that the manuscript in its current version should be rejected. In the revised manuscript I recommend omitting the quantitative future projections of CaCO3 accretion dissolution in coral reefs up to 2100, and providing a more detailed description of the experimental system

Experimental design

see above

Validity of the findings

see above

Additional comments

see above

Reviewer 3 ·

Basic reporting

The article is well written, concise and straightforward. The authors describe background material clearly and weave it nicely with their experiment. The experiment is well designed and ‘self-contained’. The findings they report are of extreme importance to reef conservation efforts worldwide. In addition to gaining a deeper understanding of reef formation and the threats it is facing.

Experimental design

The authors ask specific questions “Whether perforate and imperforate corals” will dissolve differently under estimated pH decreases for oceans and how that impacts the accretion of reefs putting into account the expected increases in the water level as well”

They conduct the experiment using a simple technique, which is measuring the dry weight of the coral skeleton after being exposed to different pH conditions. Proper replicas were used and the data seems convincing. The method they describe can also be repeated on other corals or calcifying reef-organisms as well.

After generating the data, they employ various statistical and modeling procedures to calculate the reef accretion model. Overall the methods and techniques used are ethical and are suitable for addressing the question they are asking.

Validity of the findings

The data generated, is robust, interesting and conclusive. They show that perforate corals are likely to dissolve faster than imperforate corals. This is of great importance to reef biologists and conservation efforts, and while the idea seems almost “common sense”, common sense is not quite common. In addition, by integrating their data into a model, they add another layer of significance since they find out that not only by dissolving faster, it is likely that they will be gone first. But, the whole reef won’t grow fast enough to catch up with the predicted sea-level rise. The authors are quite careful to mention that these results reflect reef accretion and not necessarily how the corals themselves are dealing with the acidification, thus providing a new insight in that perspective.

Additional comments

The GPS coordinates of the collected specimens should be added to the text

I highly recommend that the authors publish the Matlab scripts with their manuscript in order to allow other scientists to replicate their findings.

---

## Round 0.2 · accepted · Accept

Thank you for your thoughtful responses.